# Evaluating the Performance of Integrated Management of Acute Malnutrition Programs in Somalia: A Systematic Review and Meta-Analysis

**DOI:** 10.3390/ijerph22030378

**Published:** 2025-03-05

**Authors:** Andre M. N. Renzaho, Chandrakala Jaiswal, Annastancia Chineka, Musdafa Omar Aden, Abdikadir Dahir, Hanad Abdi Kari, Simon Karanja, Ajwang Fatuma, Bashir Abdi Shire, Kh Shafiur Rahaman, Mohamed Isse Mohamed, Farhan Mohamed, Nejmudin Kedir Bilal, Gabriel Ocom, Mohamed Ag Ayoya, Biram Ndiaye, Eric Alain Ategbo

**Affiliations:** 1School of Medicine, Translational Health Research Institute, Western Sydney University, Locked Bag 1797, Penrith, NSW 2571, Australia; shafiur.rahaman@westernsydney.edu.au; 2UNICEF Somalia, API Compound, Nairobi P.O. Box 44145, Kenya; cjaiswal@unicef.org (C.J.); mouaden@unicef.org (M.O.A.); abdahir@unicef.org (A.D.); hkarie@unicef.org (H.A.K.); skaranja@unicef.org (S.K.); afatuma@unicef.org (A.F.); bashire@unicef.org (B.A.S.); mimohamoud@unicef.org (M.I.M.); nkbilal@unicef.org (N.K.B.); gocom@unicef.org (G.O.); mayoya@unicef.org (M.A.A.); eaategbo@unicef.org (E.A.A.); 3Somalian Ministry of Health, Corso Somalia, Mogadishu MC 13-1302, Somalia; nutrition@moh.gov.so; 4World Bank, 1818 H Street, Washington, DC 20433, USA; biram.ndiaye@yahoo.fr

**Keywords:** integrated management of acute malnutrition wasting, SAM, OTP, community-based management of acute malnutrition, IMAM, scaling-up, Somalia

## Abstract

**Background**: Globally, acute malnutrition remains a significant public health challenge. Severe acute malnutrition (SAM) is the most lethal type of acute malnutrition. This study aimed to produce pooled estimates of the effectiveness of integrated management of acute malnutrition (IMAM) programs in addressing SAM in Somalia. **Methods**: Medline, Embase, Cochrane, Web of Science, CINAHL, ProQuest, Google Scholar, eLENA, and the UNICEF website were searched with no language or date restrictions. Random effect models were used to estimate the pooled estimates of outpatient therapeutic program (OTP) and stabilization centres (SC) performance outcomes (I^2^ > 50%). **Results**: Of 186 identified studies, three included data from IMAM programs in Somalia but were excluded, as they had incomplete data. Included were seven datasets identified through the authors’ list, which screened 36.34 million and admitted 2.31 million (6.35%) children aged 6–59 months. The pooled estimates of IMAM performance outcomes [%, (95% confidence interval)] for OTPs and SCs were, respectively, 2.45 (2.18, 2.56) and 4.11 (95%CI: 3.33, 4.90) for relapse, 95.39 (94.87, 95.90) and 80.81 (79.25, 82.37) for recovery, 0.18 (0.15, 0.21) and 1.73 (1.51, 1.95) for death, 2.57 (2.34, 2.79) and 2.75 (2.37, 3.12) for defaulter, 1.86 (1.55, 2.17), and 0.84 (0.66, 1.02) for non-recovery. **Conclusions**: OTP and SC performance outcomes in Somalia exceeded the Sphere Minimum Standards and compare favourably with results from the region. The adaptation of IMAM programs to meet multiple challenges in Somalia, supported by well-designed, coordinated, standardized, integrated, and harmonized implementation plans, is a strength.

## 1. Introduction

Child malnutrition, which encompasses foetal growth restriction, stunting, severe acute malnutrition, and deficiencies of vitamin A and zinc, along with suboptimal breastfeeding, continues to account for 45% of all deaths among children under five years of age [1]. Ensuring adequate nutrition early in life is a global priority and has been a key focus of international agendas promoting health and productivity. The Scaling Up Nutrition (SUN) Movement seeks to eliminate malnutrition in all its forms by 2030. The Sustainable Development Goals (SDGs)—Goal 2: Zero Hunger and Target 2.2.—pledge the following: “*by 2030, end all forms of malnutrition, including achieving, by 2025, the internationally agreed targets on stunting and wasting in children under 5 years of age, and address the nutritional needs of adolescent girls, pregnant and lactating women and older persons*”. Despite global efforts, data from the Global Burden of Disease indicate that child growth failure and nutritional deficiencies still pose significant public health challenges, causing the largest number of deaths among children under the age of five in low- and middle-income countries [2]. The recent 2023 SDGs Special Report [3] found that the number of people experiencing hunger (i.e., <1800 calories/day) and food insecurity (i.e., the lack of or inadequate regular access to enough safe and nutritious food for normal growth, development, and an active and healthy life) has been rising since 2015. By 2022, about 9.2% of the world population (735 million people) experienced chronic hunger, about 29.6% (2.4 billion people) reported moderate or severe food insecurity, whilst 45 million children under the age of five suffered from acute malnutrition (weight-for-height <−2 Z scores and/or bilateral oedema), and 148 million were stunted (height-for-age <−2 Z scores). Progress in tackling all forms of malnutrition has remained unacceptably slow. Data by the Independent Expert Group summarized in the 2022 Global Nutrition Report [4] found that less than 50% of countries are expected to meet at least one of the nine global nutrition targets, but none are expected to achieve all the nine targets. Only five countries are on track to meet four targets. Although the world’s initial SDG gap is to be closed by 2030, the world has alleviated only 44% and 33% of the burden of stunting and severe acute malnutrition, respectively. Hence, based on the current pace of advancement, less than half of the countries in South Asia and Africa will achieve the 2030 SDG goals.

One of the life-saving interventions widely used to meet global nutrition targets is the scale-up of the integrated management of acute malnutrition (IMAM) model. Before the uptake of the IMAM model, all three clinical outcomes for severe acute malnutrition (SAM), namely marasmus (weight-for-height <−3 Z scores or <70% of the median), bilateral pitting oedema regardless of anthropometric outcomes (known as kwashiorkor or oedematous malnutrition), and marasmic kwashiorkor (weight-for-height <−3 Z scores and bilateral oedema), were treated in therapeutic feeding centres, paediatric wards, or nutrition rehabilitation units. The costs of these facilities were high due to their significant resource needs, skilled personnel, and the exportation of therapeutic goods [5]. Parents were required to accompany children admitted to the program and be away from the rest of the family. The programs had low coverage rates characterized by late presentation, medical complications overcrowding, and an increased risk of cross-infections [6].

Due to these limitations, the IMAM model, initially piloted in 2000, evaluated, and endorsed by the United Nations in 2007 was seen as the best practice. It has been scaled up to become the norm [7]. Since then, the IMAM model has gained widespread acceptance and popularity in humanitarian and non-humanitarian contexts across low- and middle-income countries [7]. The IMAM model has three components: dry take-home supplementary foods to treat children with moderate acute malnutrition without medical complications, ready-to-use therapeutic foods (RUTFs) to use in outpatient therapeutic programs (OTPs) to treat children with SAM without medical complications, and inpatient care or stabilization centres (SCs) to treat children with severe or moderate malnutrition with medical complications [8]. However, current practices in Somalian SCs do not accommodate moderately malnourished children with medical complications. Globally, IMAM programs are cost-effective when compared with other priority health interventions [9,10].

Somalia, one of the 66 SUN member states, has not had robust documentation of success stories. Data presented in this paper focus on the treatment of SAM in Somalia for several reasons. SAM remains the most visible and lethal type of acute malnutrition in the country, and children with SAM are twice as likely to die than those with moderate acute malnutrition [11]. SAM remains one of the top threats to child survival, with excess mortality among severely malnourished children skyrocketing when acute malnutrition combines with disease outbreaks [11]. The likelihood of SAM increases within an impaired social and health environment and is magnified by underlying problems such as poverty, natural and human-made disasters such as famine or war, inadequate quantity and quality of food, and infection and disease outbreaks [12]. Over decades, Somalia has had infectious disease outbreaks, recorded high levels of food insecurity, and experienced a continuous cyclic poly-crisis stemming from complex social and political dynamics [13]. Consequently, Somali children’s growth has been affected by shifting heterogenous hyperpolarized conflict and political instability, compounded by climate-driven extreme crises (e.g., floods, crippling droughts, and famine), mass displacement, and inadequate health and social care [13]. The country has developed guidelines that govern the implementation and evaluation of IMAM programs, and progress has been made to scale up effective interventions to treat SAM [14,15], supported by internationally endorsed minimum standards (The SPHERE guidelines) [16]. However, multi-country case studies and systematic review studies examining the effectiveness of OTPs and SCs in treating SAM [16,17,18,19,20,21,22,23,24,25] have mainly included data from Mali, Malawi, the Democratic Republic of Congo, Kenya, Niger, Senegal, Sudan, Burkina Faso, Zambia, Ethiopia, Sierra Leonne, and Cameroon. None of them have included data from Somalia due to the paucity of data stemming from poor knowledge mobilization and dissemination strategies.

Therefore, little is known about the effectiveness of OTPs and SCs in Somalia and the quality of recovery of children admitted with SAM. This study aimed to produce pooled estimates of the effectiveness of OTPs and SCs in addressing SAM in Somalia and compare them with the SPHERE minimum standards. The primary research question was this: (1) What is the effectiveness of OTPs and SCs in treating SAM among children aged 6–59 months in Somalia? To answer this question, the performance of OTPs and SCs in improving the quality of care and treatment outcomes of children 6–59 months with SAM was compared with international minimum humanitarian standards [16].

## 2. Materials and Methods

This study was conducted according to the preferred reporting items for systematic review and meta-analysis [26].

### 2.1. Source of Information

A search of the following databases was conducted during July 2024 with no restriction applied to language or date: Medline, Embase, Cochrane, Web of Science, CINAHL, ProQuest, Google Scholar (screening the first 200 search records), eLENA (WHO), and the UNICEF website. The reference lists of included studies were also screened, and authors were contacted to gain access to raw data. No restriction was applied to language or date to reflect a historical perspective. The systematic review was registered by the PROSPERO International Prospective Registry for systematic reviews (reference number CRD42024569995).

### 2.2. Search Strategy

Two authors (SR and AR) independently searched bibliographical databases, grey literature, and various organizations’ websites using search terms based on MeSH or subject heading truncations (*) and Boolean operators using the following combination: “malnutrition” AND “integrated management of acute malnutrition” and “children under 5 years” and “Somalia”, extended as shown below. Key terms for organizations’ websites and Google Scholar were verified before the search and amended as appropriate.

**Malnutrition**[(“malnutrition” [MeSH] OR malnourish * OR undernutrition OR wasting OR “Acute Malnutrition” [MeSH] OR MAM OR SAM or undernutrition OR severe acute malnutrition or moderate acute malnutrition or “Protein-Energy Malnutrition” [MeSH] OR under-nutrition OR underweight OR wast * OR “weight for height” OR “weight-for-height” OR “weight for length” OR “weight-for-length” OR “mid-upper arm circumference” OR“mid-upper arm circumference” OR MUAC OR “wasting syndrome” [MeSH] “infant nutrition disorders” [MeSH])] **AND****Integrated management of acute malnutrition**[ “ready to use therapeutic food * OR RUTF * OR F100 OR F75 OR F-100 or F-75 or CTC OR Community-based Therapeutic Care or FBF or “fortified blended foods” OR IMAM OR “integrated management of acute malnutrition” OR “integrated community case management” OR “inpatient management” OR “in-patient management” OR “facility based management” OR “facility based management” or WHO Protocol or World Health Organisation protocol or “Food” [MeSH] OR “infant food” [MeSH] OR “food, fortified” [MeSH] OR “food, formulated” [MeSH] OR “dietary supplements” [MeSH] OR “dietary fat *” [MeSH] OR “Milk Proteins” [Mesh] OR “fortified food *” OR plumpynut or Plumpy’Nut^®^ or Plumpy’Nut OR “dietary protein *” OR “corn soy” OR “Wheat soy * blend *” OR “Rice milk blend *” OR “Milk rice blend *” OR “Pea wheat blend *” OR “Cereal pulse blend *” OR “Lipid-based nutrient supplement *” OR Nutributter OR “lipid based nutrient supplement *]**AND****Children < 5 years**[under five” OR kid * OR kids paediatr * OR pediatr * OR child * or “Infant” [MeSH] OR “Child, preschool” [MeSH] OR Infant * OR toddler * OR baby OR babies OR preschool OR newborn * OR neonate * OR kindergarten OR under-5 * OR “under 5 *” OR under-five]**AND****Country**[Somalia]

### 2.3. Inclusion and Exclusion Criteria

The inclusion criteria were informed by the PICOTS framework [27] as follows:**Population**: children with SAM aged 6–59 months;**Intervention**: IMAM, mainly OTP and SC **cohorts**;**Comparison**: cohort monitoring through pre (admission)-post (discharge) analysis;**Outcome**: primary: recovery rate, death rate, defaulter rate, and non-recovery rate; secondary: relapse rate, weight gain velocity, medical transfer rate, and length of stay in the program;**Timing**: no date restriction for the search;**Setting**: Somalia.

Only studies on Somalia were included in this review due to the country’s unique nutrition situation: food insecurity remains very high despite long-term humanitarian, development, and peace interventions [13]. The latest data suggest that over the last three decades, food insecurity has remained above Integrated Food Security Phase Classification (IPC) ‘Crisis’ levels (Phase 3+). As of March 2024, approximately 17% of the population (3.2 million people) are in IPC Phase 3 (crisis) and around 4% of the population (800,000 people) were experiencing worse conditions—IPC Phase 4 (emergency), with a projected 1.7 million children aged 6–59 months projected to face severe acute malnutrition by December 2024 (430,000 with severe acute malnutrition) [28]. Over decades, the overall acute malnutrition prevalence has consistently averaged the 15% emergency threshold [29]. Studies included focused on evaluating the effectiveness of IMAM programs in Somalia. They included studies summarizing data from inpatient care or SCs and OTPs implemented through health posts or health centres. Included studies or data monitored admitted children over time using pre-tests for admitting children as the baseline and clearly defined exit criteria to discharge children as defined by the Sphere Minimum Standards [30]. Studies that focused on evaluating the effectiveness of alternative feeding products other than prescribed those mandated by the Sphere Minimum Standards [16] through experimental or quasi-experimental designs were excluded. These included studies trialling the utility and efficacy of specially formulated lipid-based nutrient supplements or milk-free ready-to-use therapeutic food made from soya, maize, and sorghum. Other excluded studies included those that solely focused on growth monitoring or on adults, carried out outside sub-Saharan Africa or in African countries other than Somalia. Review papers, conference presentations, protocols, or incomplete short reports were also excluded.

### 2.4. Data Extraction and Quality Assessment

All retrieved studies were exported to the EndNote 20 reference manager to identify duplicates and manage the screening process. After removing duplicates, retrieved studies were screened by title and abstract to assess their relevance to the study purpose. Documents retained after the screening of abstracts were subjected to a full reading of their text to determine their eligibility. Both authors read through the reference lists of the retained documents to identify any further relevant studies and identify authors to be contacted to gain access to the raw data. The search, extraction, and quality appraisal of documents and datasets were undertaken by one author and independently verified by a second author. Data extraction was carried out using a modified form of the Joanna Briggs Institute data extraction form [31] to capture key information on the characteristics, and for raw data, thirteen data quality assessment items were used to establish the trustworthiness and robustness of included datasets [32]. Each item for data quality was scored using a 3-point Likert scale as follows: 2 for “Yes, completely”, 1 for “Partly”, and 0 for “No, not at all”, with a score ranging from 0 to 26. Agreement between the two authors extracting and assessing the quality of the datasets was assessed using weighted Kappa coefficients using quadratic weights. Begg’s and Egger’s tests were used to assess publication bias and small-study effects (funnel plots and regression- and correlation-based methods) [33].

### 2.5. Outcomes of Interest

The main outcomes of interest were indicators used to evaluate the effectiveness of OTPs and SCs as defined in the Sphere Humanitarian Charter and Minimum Standards [30] and included the following:

Primary outcomes:


**Treatment outcomes**


**Recovered rate**: children discharged after a successful recovery. It is calculated as the number of children recovered/total number of discharged × 100.**Death rate:** children who died during treatment in the nutrition programs. It is calculated as the number of deaths/total number of discharged × 100.**Defaulter rate**: children who did not complete treatment due to absenteeism (absent during three consecutive visits, defaulter confirmed at third absence). It is calculated as the number of defaulters/total number of discharged × 100.


**Quality of care and treatment outcomes indicators:**
**Not recovered/cured**: children who did not meet the discharge criteria for recovery after four months of treatment. It is calculated as the number of individuals not recovered/total number of discharged × 100.**Relapse rate:** children who completed treatment and were discharged as “recovered” but developed wasting within a period of two months and were readmitted for further treatment. It is calculated as the number of re-admission/total admission × 100.**Weight gain velocity**: calculated as weight gain (weight at discharge−weight at admission in grams)/(the weight on admission in kilograms × length of stay in the program); expressed as g/kg/person/day.**Length of stay**: calculated as the date at discharge minus the date at admission, expressed in days.**Medical transfers**: children referred to a hospital or health facility outside nutritional programs for further medical investigation or treatment. It is calculated as the number of medical transfers/total number of discharged × 100.


However, data made available to the research team were aggregated and did not contain indicators related to weight gain velocity and length of stay in the feeding program.

### 2.6. Data Analysis

Data were analyzed using STATA 18 (Stata Corporation, College Station, TX, USA). We computed pooled estimates of outcomes and their 95% confidence intervals by sex. We assessed heterogeneity using a forest plot and I^2^ test. Random effect models were used in the analyses for a high level of heterogeneity (I^2^ ≥ 50%). Egger’s tests were used for assessing publication bias and small-study effects (funnel plots and regression- and correlation-based methods) [33]. The STATA command “metabias” was used to test for funnel plot asymmetry. Egger’s regressions were used to check for asymmetry in a funnel plot; regression-based methods are useful for publication bias, especially when detecting small-study effects [34,35]. Funnel plots, weighted correlation, Egger’s regression bias coefficients, and regression scatter plots were produced for each study outcome (Appendix A). A non-statistically significant Egger’s regression bias coefficient provided weak evidence for the presence of small-study effects [34].

## 3. Results

### 3.1. Characteristics of Included Studies

A total of 198 studies (184 peer-reviewed studies, two reports, and seven datasets) were identified. The two reports were excluded, as one was an incomplete non-peer-reviewed report and the other was an IMAM policy guideline. For the peer-reviewed papers, 137 records were removed before screening (duplicates) and a further 44 records were excluded after screening them for relevance based on title and abstract. The remaining three studies included data from IMAM programs in Somalia and were screened for relevance based on full text. The study conducted by Shragai and colleagues [36] included 12 OTP facilities and focused more on IMAM programs’ adaptation to the COVID-19 pandemic. Hence, it was excluded from our study. Ngoy and colleagues [37] provided a short report on one pediatric inpatient care centre at Istarlin Hospital in the Guriel district. The facility had limited data and was located in a conflict-affected region at the time of the study; therefore, it was subsequently excluded. Ntambi and colleagues [38] undertook a bottleneck analysis for IMAM programs without providing an analysis of the programs’ performance (Figure 1). Screening of references of retrieved studies identified UNICEF Somalia as the custodian of routine aggregated data on the performance of IMAM programs, which were publicly available on request. We contacted UNICEF Somalia and were granted access. Therefore, the current systematic review included **seven reports containing cohort monitoring data among 6-to-59-month-old children from 1 January 2018 to 30 June 2024**. Over the 6.5 years, 36,341,666 (47.1% boys and 52.9% girls) children aged 6–59 months were screened, of whom 2,308,873 (44.6% boys and 55.4% girls) admitted to feeding centres (Table 1).

### 3.2. OTP and SC Admissions

The number of children admitted to OTPs and SCs increased exponentially over the study period. A total of 2,308,873 6-to-59-month-old children were admitted to feeding programs over 6.5 years, of whom 2,147,668 (93%) were admitted to OTPs and 161,205 (7%) were admitted to SCs. Of children admitted to OTPs and SCs, 82.7% (or 1,776,213) and 97.5% (or 157,192) exited the programs, respectively. The numbers of children who recovered, died, defaulted, and did not recover are summarized in Table 2. The data quality scores ranged from 18 to 23 out of a total score of 26, suggesting high-quality datasets. Kappa statistics and percentage of agreement ranged from 0.758 to 0.916 and 0.942 to 0.981 respectively, suggesting excellent agreement.

### 3.3. Performance of OTPs

Forest plots depicting the performance of OTPs and heterogeneity statistics are summarized in Appendix A. As can be seen from Table 3, the pooled re-admission/relapse rate after the cure was 2.45% (95% CI: 2.32, 2.58). The pooled recovery rate was estimated at 95.39% (95%CI: 94.87, 95.90), whereas death, defaulter, and non-recovery rates were 0.18% (95% CI: 0.15, 0.21), 2.57% (95% CI: 2.34, 2.79), and 1.86% (95% CI: 1.55, 2.17), respectively. The performance of OTPs did not differ by sex.

### 3.4. Performance of SCs

Heterogeneity statistics and forest plots depicting the performance of SCs are summarized in Appendix A. As can be seen from Table 4, the pooled re-admission/relapse rate after cure was 3.77% (95%CI: 3.25, 4.29). The pooled recovery rate was estimated at 80.81% (95%CI: 79.25, 82.37), whereas death, defaulter, and non-recovery rates were 1.73% (95%CI:1.51, 1.95), 2.75% (95%CI: 2.37, 3.12), and 0.84% (95%CI: 0.66, 1.02), respectively. The pooled transfer rate was 13.81% (95%CI: 12.50, 15.11). The performance outcomes of SCs did not differ by sex.

### 3.5. Publication Bias

Publication year and small-study effects were found to be the potential source of heterogeneity for all performance outcomes, except for relapse and death rates in SCs (Appendix A).

## 4. Discussion

Our study sought to estimate the treatment outcomes of 6-to-59-month-old children with SAM admitted to OTPs and SCs in Somalia. IMAM programs in Somalia achieved recovery, death, defaulter, and non-recovery rates of 95.4% and 80.8%, 0.2% and 1.7%, 2.6% and 2.8%, and 1.9% and 0.8% for OTP and SC programs, respectively. Given that the IMAM program’s primary objective is to maximize child survival and reduce fatality rates among children, the observed death rates of <2% are a welcome and timely finding, especially as Somalia has experienced continuous cyclic infectious disease outbreaks, high levels of food insecurity, and cyclic polycrisis [13]. Overall, the observed OTP and SC performance outcomes compare favourably with the Sphere Minimum Standards for recovery (>75%), death (<10%), and default (<15%) rates. Our results also compare favourably with results from other African regions [17,18,19,21,22,37,38,39,40,41,42,43]. Data from Ethiopia and sub-Saharan Africa have reported recovery, death, defaulter, and non-recovery rates of 70–73%, 2–10%, 10%, and 5–15%, respectively [39,40,41,42,43]. The pooled recovery rate for sub-Saharan Africa has remained low and averaged 71.2% [21]. These figures compare unfavourably with our findings. The main challenges in these sub-regions have included non-adherence to treatment guidelines, presence of co-morbidities and untreated recurrent infections, late presentation, inadequate discharge criteria, and compromised access to RUTFs (e.g., food sharing, trading of RUTFs either to meet other needs or because of disliking the taste) [18,40]. The lack of efficiency in healthcare facilities is another major issue. A systematic review assessing the efficiency of healthcare services in sub-Saharan Africa found that less than 40% of the studied facilities were efficient [19]. The same study showed that the inefficiency of health facilities was intricately linked to the catchment population, facility ownership, and location.

Nonetheless, our findings are similar to those reported in South Sudan [6], Niger [44], and elsewhere [45]. A multi-country case study in sub-Saharan Africa found that, despite countries having different motivations for initiating IMAM programs, those that have been successful in addressing severe acute malnutrition have integrated IMAM into their essential health packages and policy and strategic documents [20]. Enabling factors include strong leadership by the ministries of health supported by key donor and implementing partners to extend service coverage and improve treatment outcomes, the decentralization of community mobilization and OTP services, and the use of robust triage criteria to identify children with SAM to be admitted to OTPs and SCs [20,22]. These factors have been the cornerstone of the success of IMAM programs in Somalia [14,15], against the backdrop of increased mobility and its impact on Somali resilience capacity, a political economy in which the diversion of aid is systemic [46].

There have been several positive signs of progress that could support our findings. Our observed low relapse rate suggests good linkage and referral systems between OTPs, SCs, and supplementary feeding programs, as well as optimal care practices and admission and exit criteria [6]. In addition, the humanitarian system has adapted well to the unique changing Somali context by supporting fixed and mobile OTPs [14,15,47]. Unfortunately, analysis by types of OTP (mobile vs. fixed) programs was challenging due to incomplete aggregated data. Nonetheless, mobile OTPs allow the humanitarian response to reach people who are marginalized and excluded from accessing much-needed aid and treatment. First piloted in 2004 to respond to the needs of Somalis severely affected by drought and the measles epidemic, mobile health teams had neither formal training nor specific guidelines to support their work [14,15,47]. Over the last decades, mobile health teams have evolved and expanded to mobile health and nutrition teams to include IMAM, supported by training packages and guidelines that address three components: maternal and newborn care services (e.g., ante and post-natal services), child health (e.g., IMAM), and capacity building (e.g., support in disease outbreak response, health and nutrition service delivery, community mobilization and active case finding and referral, and system strengthening) [47].

Therefore, mobile OTPs have strengthened disease surveillance in hard-to-reach areas and established robust systems to identify risk factors and facilitate early case detection of malnutrition and referral pathways for treatment. They have played a critical role in the lifesaving of vulnerable communities who are on the margin. The mobile health and nutrition team service implementation guideline [47] was developed to standardize and harmonize mobile health and nutrition service packages across government departments and IMAM implementing partners, articulate start-up and planning steps for IMAM implementing partners, and set standards related to minimum service requirements and operational plans. These measures have translated into increased coverage of children admitted to IMAM, increased geographical access to high-quality, simplified, and sustainable malnutrition treatment pathways, and strengthened the resilience of health services to adequately address child malnutrition. Consequently, international aid actors in Somalia have been working in more integrated ways across development, humanitarian, peace- and state-building, and have embraced programming adaptations that better support mobility to build resilient livelihoods [46].

## 5. Strengths and Limitations

The findings of our study need to be interpreted in the context of some important strengths and limitations. Strengths include the use of a robust study design supported by an internationally mandated framework to guide the conduct of systematic reviews, the inclusion of comprehensive nationally representative data, the application of rigorous analytical approaches, and the use of a politically volatile country such as Somalia as a case study. However, there are a number of limitations worth outlining. The analyzed data included one aggregate age bracket (6–59 months) and did not include children aged < 6 months. The effectiveness of OTPs and SCs by child age could not be tested. The datasets did not contain data on the length of stay in the program and weight gain velocity. The recovery rate in uncomplicated SAM is significantly associated with weight gain, with the average time to recovery from SAM relatively longer among children with lower admission weight [48]. Children admitted with higher weight gain are more likely to survive before exiting OTPs and SCs [49]. The average weight gain is strongly influenced by anthropometric outcomes on admission and the length of stay in the program [42]. We were not able to test the independent effect of admission anthropometric outcomes, weight gain, and length of stay on recovery rates and child survival.

## 6. Conclusions

The overall performance outcomes of OTPs and SCs in Somalia surpassed the Sphere Minimum Standards and were favourable compared to outcomes in other African regions and elsewhere, as per this meta-analysis. The adaptation of IMAM programs to meet multiple challenges in Somalia, supported by well-designed, coordinated, standardized, integrated, and harmonized implementation plans, is a strength. International aid agencies and policymakers need to systematically document factors that have driven this success and assess how such lessons can be replicated and scaled up regionally.

## Figures and Tables

**Figure 1 ijerph-22-00378-f001:**
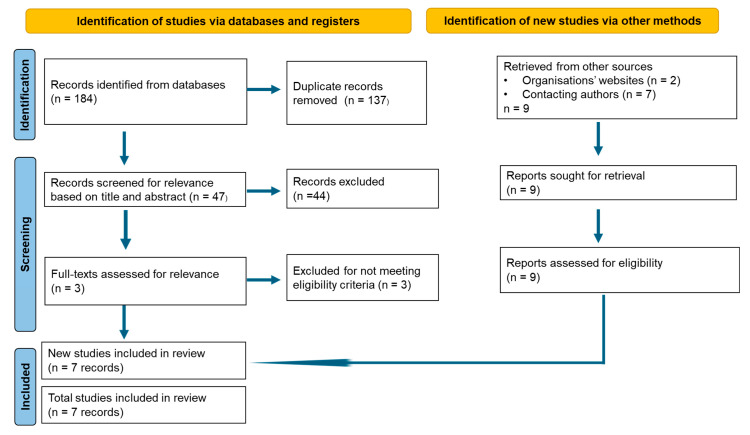
Flow chart of study selection.

**Table 1 ijerph-22-00378-t001:** Sample size and admission by program type.

Parameters	UNICEF (2018)	UNICEF (2019)	UNICEF (2020)	UNICEF (2021)	UNICEF (2022)	UNICEF (2023)	UNICEF (2024)	All Years
**Sample size (N)**								
Boys	821,918	1,765,145	2,194,856	2,520,363	3,557,944	4,547,128	1,703,926	17,111,280
Girls	894,603	1,992,753	2,419,730	2,681,427	4,048,237	5,194,678	1,998,958	19,230,386
All	1,716,521	3,757,898	4,614,586	5,201,790	7,606,181	9,741,806	3,702,884	36,341,666
**All admissions (N)**								
Boys	70,805	121,098	119,502	116,747	212,314	276,979	111,990	1,029,435
Girls	85,700	141,677	144,512	143,808	257,271	355,758	150,712	1,279,438
All	156,505	262,775	264,014	260,555	469,585	632,737	262,702	2,308,873
**OTP**								
**Admissions**								
Boys	66,595	111,328	109,239	106,459	195,446	256,907	103,700	949,674
Girls	81,243	131,935	133,972	133,639	240,649	334,996	141,560	1,197,994
All	147,838	243,263	243,211	240,098	436,095	591,903	245,260	2,147,668
**Readmission**								
Boys	2083	2682	2549	2783	4856	6059	2563	23,575
Girls	2600	3161	2869	3086	5214	7094	3235	27,259
All	4683	5843	5418	5869	10,070	13,153	5798	50,834
**Total exit**								
Boys	58,424	87,841	93,884	88,353	145,723	218,625	93,611	786,461
Girls	71,663	104,049	115,508	110,391	179,689	283,029	125,423	989,752
All	130,087	191,890	209,392	198,744	325,412	501,654	219,034	1,776,213
**SCs**								
**Admissions**								
Boys	4210	9770	10,263	10,288	16,868	20,072	8290	79,761
Girls	4457	9742	10,540	10,169	16,622	20,762	9152	81,444
All	8667	19,512	20,803	20,457	33,490	40,834	17,442	161,205
**Readmission**								
Boys	96	260	274	403	635	997	307	2972
Girls	217	276	279	457	700	1061	428	3418
All	313	536	553	860	1335	2058	735	6390
**Total exit**								
Boys	4148	9326	10,273	9951	16,237	19,723	8235	77,893
Girls	4410	9237	10,580	9790	15,891	20,287	9104	79,299
All	8558	18,563	20,853	19,741	32,128	40,010	17,339	157,192

**Table 2 ijerph-22-00378-t002:** Exits by program type among 6-to-59-month-old children.

Indicator	UNICEF (2018)	UNICEF (2019)	UNICEF (2020)	UNICEF (2021)	UNICEF (2022)	UNICEF (2023)	UNICEF (2024)	All Years
**OTPs**								
**Recovered/Cured(N)**								
Boys	55,412	81,947	89,680	84,883	139,132	211,557	89,813	752,424
Girls	67,922	97,249	110,023	105,911	171,410	274,031	120,168	946,714
All	123,334	179,196	199,703	190,794	310,542	485,588	209,981	1,699,138
**Died (N)**								
Boys	225	145	168	151	232	232	148	1301
Girls	258	175	219	170	276	256	183	1537
All	483	320	387	321	508	488	331	2838
**Defaulters**								
Boys	1832	2731	2142	1949	3597	4291	2591	19,133
Girls	2241	3191	2708	2528	4294	5439	3634	24,035
All	4073	5922	4850	4477	7891	9730	6225	43,168
**Non-recovery/not cured**								
Boys	955	3018	1894	1370	2762	2545	1059	13,603
Girls	1242	3434	2558	1782	3709	3303	1438	17,466
All	2197	6452	4452	3152	6471	5848	2497	31,069
**SCs**								
**Recovered/Cured(N)**								
Boys	2871	7453	8564	7997	13,464	16,579	6920	63,848
Girls	3159	7452	8829	7847	12,977	17,114	7710	65,088
All	6030	14,905	17,393	15,844	26,441	33,693	14,630	128,936
**Died (N)**								
Boys	61	177	201	161	321	236	126	1283
Girls	69	202	209	188	367	233	134	1402
All	130	379	410	349	688	469	260	2685
**Defaulters**								
Boys	137	316	306	366	381	371	150	2027
Girls	129	328	340	338	398	348	176	2057
All	266	644	646	704	779	719	326	4084
**Non-recovery/not cured**								
Boys	44	91	75	44	95	91	183	623
Girls	61	70	81	24	84	91	150	561
All	105	161	156	68	179	182	333	1184
**Transfers**								
Boys	1035	1289	1127	1383	1976	2446	856	10,112
Girls	992	1185	1121	1393	2065	2501	934	10,191
All	2027	2474	2248	2776	4041	4947	1790	20,303

**Table 3 ijerph-22-00378-t003:** Performance of OTPs among 6-to-59-month-old children.

Outcome	Effect Size	95% CI		% Weight
**Readmission/relapse rate**				
Boys				
UNICEF (2018)	3.13	3.00	3.26	6.86
UNICEF (2019)	2.41	2.32	2.50	7.12
UNICEF (2020)	2.33	2.24	2.42	7.13
UNICEF (2021)	2.61	2.52	2.71	7.09
UNICEF (2022)	2.48	2.42	2.55	7.22
UNICEF (2023)	2.36	2.30	2.42	7.26
UNICEF (2024)	2.47	2.38	2.57	7.10
**Random pooled effect size**	**2.54**	**2.39**	**2.68**	**49.79**
Girls				
UNICEF (2018)	3.20	3.08	3.32	6.94
UNICEF (2019)	2.40	2.31	2.48	7.16
UNICEF (2020)	2.14	2.06	2.22	7.19
UNICEF (2021)	2.31	2.23	2.39	7.17
UNICEF (2022)	2.17	2.11	2.23	7.27
UNICEF (2023)	2.12	2.07	2.17	7.30
UNICEF (2024)	2.29	2.21	2.36	7.19
**Random pooled effect size**	**2.37**	**2.18**	**2.56**	**50.21**
Overall				
**Random pooled effect size**	**2.45**	**2.32**	**2.58**	**100.00**
**Recovery/cure rate**				
Boys				
UNICEF (2018)	94.84	94.66	95.02	7.11
UNICEF (2019)	93.29	93.12	93.45	7.12
UNICEF (2020)	95.52	95.39	95.65	7.14
UNICEF (2021)	96.07	95.94	96.20	7.14
UNICEF (2022)	95.48	95.37	95.58	7.15
UNICEF (2023)	96.77	96.69	96.84	7.16
UNICEF (2024)	95.94	95.81	96.07	7.14
**Random pooled effect size**	**95.42**	**94.65**	**96.19**	**49.98**
Girls				
UNICEF (2018)	94.78	94.61	94.94	7.12
UNICEF (2019)	93.46	93.31	93.61	7.13
UNICEF (2020)	95.25	95.13	95.37	7.15
UNICEF (2021)	95.94	95.82	96.06	7.15
UNICEF (2022)	95.39	95.29	95.49	7.16
UNICEF (2023)	96.82	96.76	96.89	7.17
UNICEF (2024)	95.81	95.70	95.92	7.15
**Random pooled effect size**	**95.35**	**94.57**	**96.13**	**50.02**
Overall				
**Random pooled effect size**	**95.39**	**94.87**	**95.90**	**100.00**
**Death rate**				
Boys				
UNICEF (2018)	0.39	0.34	0.44	6.20
UNICEF (2019)	0.17	0.14	0.19	7.14
UNICEF (2020)	0.18	0.15	0.21	7.14
UNICEF (2021)	0.17	0.14	0.20	7.13
UNICEF (2022)	0.16	0.14	0.18	7.33
UNICEF (2023)	0.11	0.09	0.12	7.48
UNICEF (2024)	0.16	0.13	0.19	7.19
**Random pooled effect size**	**0.18**	**0.14**	**0.23**	**49.62**
Girls				
UNICEF (2018)	0.36	0.32	0.41	6.49
UNICEF (2019)	0.17	0.14	0.20	7.21
UNICEF (2020)	0.19	0.17	0.22	7.20
UNICEF (2021)	0.15	0.13	0.18	7.26
UNICEF (2022)	0.15	0.14	0.17	7.39
UNICEF (2023)	0.09	0.08	0.10	7.52
UNICEF (2024)	0.15	0.13	0.17	7.31
**Random pooled effect size**	**0.18**	**0.13**	**0.22**	**50.38**
Overall				
**Random pooled effect size**	**0.18**	**0.15**	**0.21**	**100.00**
**Defaulter rate**				
Boys				
UNICEF (2018)	3.14	3.00	3.28	7.04
UNICEF (2019)	3.11	3.00	3.23	7.10
UNICEF (2020)	2.28	2.19	2.38	7.15
UNICEF (2021)	2.21	2.11	2.30	7.14
UNICEF (2022)	2.47	2.39	2.55	7.17
UNICEF (2023)	1.96	1.90	2.02	7.20
UNICEF (2024)	2.77	2.66	2.87	7.13
**Random pooled effect size**	**2.56**	**2.23**	**2.89**	**49.93**
Girls				
UNICEF (2018)	3.13	3.00	3.26	7.07
UNICEF (2019)	3.07	2.96	3.17	7.13
UNICEF (2020)	2.34	2.26	2.43	7.16
UNICEF (2021)	2.29	2.20	2.38	7.16
UNICEF (2022)	2.39	2.32	2.46	7.19
UNICEF (2023)	1.92	1.87	1.97	7.21
UNICEF (2024)	2.90	2.81	2.99	7.15
**Random pooled effect size**	**2.57**	**2.23**	**2.92**	**50.07**
Overall				
**Random pooled effect size**	**2.57**	**2.34**	**2.79**	**100.00**
**Non-recovery rate**				
Boys				
UNICEF (2018)	1.63	1.53	1.74	7.12
UNICEF (2019)	3.44	3.32	3.56	7.10
UNICEF (2020)	2.02	1.93	2.11	7.14
UNICEF (2021)	1.55	1.47	1.63	7.14
UNICEF (2022)	1.90	1.83	1.97	7.15
UNICEF (2023)	1.16	1.12	1.21	7.17
UNICEF (2024)	1.13	1.06	1.20	7.15
**Random pooled effect size**	**1.83**	**1.37**	**2.29**	**49.98**
Girls				
UNICEF (2018)	1.73	1.64	1.83	7.13
UNICEF (2019)	3.30	3.19	3.41	7.12
UNICEF (2020)	2.21	2.13	2.30	7.14
UNICEF (2021)	1.61	1.54	1.69	7.15
UNICEF (2022)	2.06	2.00	2.13	7.16
UNICEF (2023)	1.17	1.13	1.21	7.17
UNICEF (2024)	1.15	1.09	1.21	7.16
**Random pooled effect size**	**1.89**	**1.42**	**2.36**	**50.02**
Overall				
**Random pooled effect size**	**1.86**	**1.55**	**2.17**	**100.00**

**Table 4 ijerph-22-00378-t004:** Performance of SCs among 6-to-59-month-old children.

Indicators	Effect Size	95% CI		% Weight
**Readmission/relapse rate**				
Boys				
UNICEF (2018)	2.28	1.85	2.78	7.03
UNICEF (2019)	2.66	2.35	2.99	7.22
UNICEF (2020)	2.67	2.37	2.98	7.23
UNICEF (2021)	3.92	3.55	4.31	7.15
UNICEF (2022)	3.76	3.48	4.06	7.26
UNICEF (2023)	4.97	4.67	5.28	7.24
UNICEF (2024)	3.7	3.31	4.13	7.1
**Random pooled effect size**	**3.43**	**2.72**	**4.14**	**50.22**
Girls				
UNICEF (2018)	4.87	4.26	5.54	6.69
UNICEF (2019)	2.83	2.51	3.18	7.21
UNICEF (2020)	2.65	2.35	2.97	7.24
UNICEF (2021)	4.49	4.1	4.91	7.11
UNICEF (2022)	4.21	3.91	4.53	7.24
UNICEF (2023)	5.11	4.81	5.42	7.24
UNICEF (2024)	4.68	4.25	5.13	7.06
**Random pooled effect size**	**4.11**	**3.33**	**4.9**	**49.78**
**Overall**				
**Random pooled effect size**	**3.77**	**3.25**	**4.29**	**100**
**Recovery/cure rate**				
Boys				
UNICEF (2018)	69.21	67.78	70.6	6.88
UNICEF (2019)	79.92	79.09	80.7	7.15
UNICEF (2020)	83.36	82.63	84.1	7.18
UNICEF (2021)	80.36	79.57	81.1	7.16
UNICEF (2022)	82.92	82.33	83.5	7.22
UNICEF (2023)	84.06	83.54	84.6	7.23
UNICEF (2024)	84.03	83.22	84.8	7.16
**Random pooled effect size**	**80.61**	**78.2**	**83**	**49.97**
Girls				
UNICEF (2018)	71.63	70.28	73	6.92
UNICEF (2019)	80.68	79.86	81.5	7.15
UNICEF (2020)	83.45	82.73	84.2	7.18
UNICEF (2021)	80.15	79.35	80.9	7.16
UNICEF (2022)	81.66	81.05	82.3	7.21
UNICEF (2023)	84.36	83.85	84.9	7.23
UNICEF (2024)	84.69	83.93	85.4	7.17
**Random pooled effect size**	**81.00**	**78.76**	**83.2**	**50.03**
Overall				
**Random pooled effect size**	**80.81**	**79.25**	**82.4**	**100**
**Death rate**				
Boys				
UNICEF (2018)	1.47	1.13	1.89	6.51
UNICEF (2019)	1.9	1.63	2.2	7.06
UNICEF (2020)	1.96	1.7	2.24	7.11
UNICEF (2021)	1.62	1.38	1.89	7.23
UNICEF (2022)	1.98	1.77	2.2	7.4
UNICEF (2023)	1.2	1.05	1.36	7.67
UNICEF (2024)	1.53	1.28	1.82	7.13
**Random pooled effect size**	**1.66**	**1.4**	**1.93**	**50.11**
Girls				
UNICEF (2018)	1.56	1.22	1.98	6.51
UNICEF (2019)	2.19	1.9	2.51	6.94
UNICEF (2020)	1.98	1.72	2.26	7.13
UNICEF (2021)	1.92	1.66	2.21	7.09
UNICEF (2022)	2.31	2.08	2.56	7.3
UNICEF (2023)	1.15	1.01	1.3	7.69
UNICEF (2024)	1.47	1.23	1.74	7.23
**Random pooled effect size**	**1.79**	**1.42**	**2.17**	**49.89**
Overall				
**Random pooled effect size**	**1.73**	**1.51**	**1.95**	**100**
**Defaulter rate**				
Boys				
UNICEF (2018)	3.3	2.78	3.89	6.54
UNICEF (2019)	3.39	3.03	3.78	7.07
UNICEF (2020)	2.98	2.66	3.33	7.17
UNICEF (2021)	3.68	3.32	4.07	7.07
UNICEF (2022)	2.35	2.12	2.59	7.37
UNICEF (2023)	1.88	1.7	2.08	7.44
UNICEF (2024)	1.82	1.54	2.13	7.26
**Random pooled effect size**	**2.75**	**2.21**	**3.3**	**49.93**
Girls				
UNICEF (2018)	2.93	2.45	3.47	6.7
UNICEF (2019)	3.55	3.18	3.95	7.05
UNICEF (2020)	3.21	2.89	3.57	7.15
UNICEF (2021)	3.45	3.1	3.83	7.09
UNICEF (2022)	2.5	2.27	2.76	7.35
UNICEF (2023)	1.72	1.54	1.9	7.46
UNICEF (2024)	1.93	1.66	2.24	7.27
**Random pooled effect size**	**2.75**	**2.17**	**3.33**	**50.07**
Overall				
**Random pooled effect size**	**2.75**	**2.37**	**3.12**	**100**
**Non-recovery rate**				
Boys				
UNICEF (2018)	1.06	0.77	1.42	6.35
UNICEF (2019)	0.98	0.79	1.2	7.13
UNICEF (2020)	0.73	0.57	0.91	7.33
UNICEF (2021)	0.44	0.32	0.59	7.49
UNICEF (2022)	0.59	0.47	0.71	7.55
UNICEF (2023)	0.46	0.37	0.57	7.63
UNICEF (2024)	2.22	1.91	2.56	6.3
**Random pooled effect size**	**0.9**	**0.61**	**1.18**	**49.78**
Girls				
UNICEF (2018)	1.38	1.06	1.77	6.1
UNICEF (2019)	0.76	0.59	0.96	7.26
UNICEF (2020)	0.77	0.61	0.95	7.32
UNICEF (2021)	0.25	0.16	0.36	7.62
UNICEF (2022)	0.53	0.42	0.65	7.57
UNICEF (2023)	0.45	0.36	0.55	7.64
UNICEF (2024)	1.65	1.4	1.93	6.72
**Random pooled effect size**	**0.8**	**0.53**	**1.06**	**50.22**
Overall				
**Random pooled effect size**	**0.84**	**0.66**	**1.02**	**100**
**Transfer rate**				
Boys				
UNICEF (2018)	24.95	23.64	26.3	6.8
UNICEF (2019)	13.82	13.13	14.5	7.16
UNICEF (2020)	10.97	10.37	11.6	7.19
UNICEF (2021)	13.9	13.22	14.6	7.17
UNICEF (2022)	12.17	11.67	12.7	7.23
UNICEF (2023)	12.4	11.94	12.9	7.24
UNICEF (2024)	10.39	9.74	11.1	7.17
**Random pooled effect size**	**14.01**	**11.98**	**16.1**	**49.96**
Girls				
UNICEF (2018)	22.49	21.27	23.8	6.86
UNICEF (2019)	12.83	12.15	13.5	7.16
UNICEF (2020)	10.6	10.02	11.2	7.2
UNICEF (2021)	14.23	13.54	14.9	7.16
UNICEF (2022)	12.99	12.48	13.5	7.22
UNICEF (2023)	12.33	11.88	12.8	7.24
UNICEF (2024)	10.26	9.64	10.9	7.19
**Random pooled effect size**	**13.62**	**11.77**	**15.5**	**50.04**
Overall				
**Random pooled effect size**	**13.81**	**12.5**	**15.1**	**100**

## Data Availability

The original contributions presented in this study are included in the article/Appendix A. Further inquiries can be directed to the corresponding author.

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
