# Peer review of "Evaluating the Performance of Integrated Management of Acute Malnutrition Programs in Somalia: A Systematic Review and Meta-Analysis"

_ijerph, 2025, doi:10.3390/ijerph22030378_

Round 1
Reviewer 1 Report
Comments and Suggestions for Authors
Reviewer Evaluation Comment:
In order to determine how well the current approaches and interventions are working to address acute malnutrition in Somalia, the research study "Evaluating the Performance of Integrated Management of Acute Malnutrition Programs in Somalia: a systematic review and meta-analysis" is important. The study can offer important insights into how to better deliver nutritional support to vulnerable populations by identifying the programs' strengths and weaknesses. Enhancing the overall effect of malnutrition management initiatives and making sure that resources are used effectively to save lives and advance improved health outcomes for Somalian families and children depend on this evaluation.
Hence, my comments below:
1. Title
Title: Evaluating the performance of integrated management of acute malnutrition programs in Somalia: A systematic re-2 view and metanalysis
The spelling metanalysis is wrong, it should be Meta-analysis. Take note of that.
The title should be: Evaluating the Performance of Integrated Management of Acute Malnutrition Programmes in Somalia: a Systematic Review and Meta-analysis.
2. Abstract
Abstract should follow this format – Background, Methods, Results and Conclusions. This will make the content of the abstract meaningful.
3. Background or Introduction
The Background do not include much vital information on the subject matter. The background of this study must follow the ‘Funnel Shape’ to scope and review existing studies across developed countries, followed by developing countries and the study setting.
The introduction is the place to review other conclusions on the topic, which include both older and modern existing studies. The background information shows that the researcher is aware of prior research, and introduces past findings to those who might not have that expertise. The introduction section should have steps such as: Introduce your topic. The first job of the introduction is to tell the reader what the topic is and why it's interesting or important; Describe the background; Establish your research problem; Specify your objective(s); and Map out the paper, as there are general phases that are associated with introduction section writing. This also include stating the study intent, outlining the key study characteristics, describing key findings from previous studies (statistical prevalence or predictors found in other studies), and giving an overview structure of the paper, which include the main objectives and the specific objectives. Conclude with an attempt for policy driven or intervention or awareness driven programmes intended for this study. Kindly revise. Some of the few potential questions that remind us of the detailed and critical questions to ponder on when re-working the ‘Introduction section’. These are: what are the steps taken in order to ensure more comprehensive data collection from IMAM programs in Somalia? Has it been fruitful? Does IMAM programmes vary across different regions within Somalia, and if it does, what impact does this have on the outcomes in Somalia?
4. Methods
Provide a description of the study design - what is the type of design you employed in this study even though it is a systematic review and meta-analysis approach you employed. Is it cross-sectional approach in the collection of existing studies or in the longitudinal form.
Give a detailed geographical description of the study setting and what is the justification for the choice of the study setting?
Put the search strategy section in tables [143-144]
Mention the total number of studies and participants included in the final analysis, if not already specified [173-186]
Was there a concern for ethical considerations? If yes, what were ethical principles you employed when retrieving these studies for the systematic review and meta-analysis? Have it in mind that ethical considerations guide your research designs and practices, concerning consent, anonymity, confidentiality (if needed such as administrative documents), risk, and results communication (very important).
5. Results
The thematic analysis is clear, but ensure each theme is introduced with a brief summary before diving into detailed findings [Line 226 - Line 314].
Consider adding specific examples or case studies to illustrate the identified themes more vividly, in the areas of performance or under-performance of integrated management of acute malnutrition programmes in Somalia.
How did you address the publication bias as mentioned in Line 311 – Line 313.
6. Discussion
The discussion section discusses the findings’ significance tied to the research objective(s), delving into the meaning, importance, and relevance of the study findings. Also, its focus on explaining and evaluating what you found must be related to the related to the existing literature review, making an argument in support or not supporting the overall argument. Therefore, let your discussion focused on the interpretation of your findings and relate to literature whether it corroborates or not, with existing studies, alongside with in-citation of recent references.
Some of these questions should come forward aligning with the study objectives you seek to fill the gap with: are there primary barriers to successfully implement IMAM programmes in Somalia, and, if no, how can this be addressed? How does the quality of care in OTPs and SCs affect the performance outcomes, and are there measures put in place for possible improvement? How do the performance outcomes of IMAM programs in Somalia compare with those in other countries facing similar challenges? Is there any trends or variations different from your findings? Can community engagement and local knowledge be integrated into the IMAM programmes to enhance their effectiveness? Any training and resources needed for healthcare providers to improve the management of acute malnutrition in Somalia? How can the long-term impact of IMAM programmes on child health and nutrition be measured and evaluated?
These will help tailor your discussion section more impactful and interesting, as these are some of the major concerns debated in High-income countries (HICs) and Low-and Middle-income countries (LMICs). Kindly revise.
7. Study Implications for future studies/research
What are the implications of your study findings? What is the contribution of this study to the existing one, especially in sub-Saharan African or in developing countries?
Regarding implications for future research, what role do national and international policies play in the success of IMAM programmes in Somalia, and how can these policies be strengthened to support IMAM programmes in Somalia?
Regarding implications for future studies, are there innovative approaches or technologies that can be incorporated into IMAM programmes to improve outcomes in Somalia?
8. Strength of this study
What is the strengths of this study? You only included the Limitations for this study. This research should have its strengths in areas of literature review, methodological approach, and findings interpretations.
9. Conclusion/Recommendations
The conclusion should stem from summing up your research paper (by the following steps - restate your research topic, restate the thesis, summarize the main points, state the significance or results, and conclude your thoughts). Always have it in mind that a conclusion is not merely a summary of the main topic(s) covered or a re-statement of your research problem, but a synthesis of key points and, if applicable, where you should recommend new areas for future research. Although, the conclusion is strong, consider reiterating the key implications for public health policy and future research.
Where is the Recommendation for this study? You can add it to the Conclusion section with the sub heading – Conclusion/Recommendations
10. Editing
Ensure consistency in tense and improve grammatical flow in some sections
11. References
Does this References of this study follow this Journal format? Kindly check and revisit them all.
Check all citations in the entire manuscript and see it corresponds with the references
Ensure all references are formatted correctly and consistently.
Comments on the Quality of English LanguageThe help of a professional editor is required and recommended.
Author Response
Dear Editor-in-chief
We appreciate the opportunity to revise and resubmit our manuscript for consideration in the International Journal of Environmental Research and Public Health. We sincerely thank you and the reviewers for the time and effort taken to provide insightful comments and suggestions. We have carefully addressed all the points raised and believe that our manuscript has improved as a result
Below we provide a point-by-point response to each comment from the reviewers. Reviewers’ comments are presented in bold, followed by our response in the regular text
Reviewer 1:
Comment 1:
In order to determine how well the current approaches and interventions are working to address acute malnutrition in Somalia, the research study "Evaluating the Performance of Integrated Management of Acute Malnutrition Programs in Somalia: a systematic review and meta-analysis" is important. The study can offer important insights into how to better deliver nutritional support to vulnerable populations by identifying the programs' strengths and weaknesses. Enhancing the overall effect of malnutrition management initiatives and making sure that resources are used effectively to save lives and advance improved health outcomes for Somalian families and children depend on this evaluation.
Response: Thank you for the complimentary comments.
Comment 2:
Title: Evaluating the performance of integrated management of acute malnutrition programs in Somalia: A systematic re-2 view and metanalysis. The spelling metanalysis is wrong, it should be Meta-analysis. Take note of that. The title should be: Evaluating the Performance of Integrated Management of Acute Malnutrition Programmes in Somalia: a Systematic Review and Meta-analysis.
Response: Thank you. done
Comment 3:
Abstract should follow this format – Background, Methods, Results and Conclusions. This will make the content of the abstract meaningful.
Response: The IJERPH states that “The abstract should be a total of about 200 words maximum. The abstract should be a single paragraph and should follow the style of structured abstracts, but without headings”. Nevertheless, these changes have been made
Comment 4:
Background or Introduction: The Background do not include much vital information on the subject matter. The background of this study must follow the ‘Funnel Shape’ to scope and review existing studies across developed countries, followed by developing countries and the study setting.
The introduction is the place to review other conclusions on the topic, which include both older and modern existing studies. The background information shows that the researcher is aware of prior research, and introduces past findings to those who might not have that expertise. The introduction section should have steps such as: Introduce your topic. The first job of the introduction is to tell the reader what the topic is and why it's interesting or important; Describe the background; Establish your research problem; Specify your objective(s); and Map out the paper, as there are general phases that are associated with introduction section writing. This also include stating the study intent, outlining the key study characteristics, describing key findings from previous studies (statistical prevalence or predictors found in other studies), and giving an overview structure of the paper, which include the main objectives and the specific objectives. Conclude with an attempt for policy driven or intervention or awareness driven programmes intended for this study. Kindly revise. Some of the few potential questions that remind us of the detailed and critical questions to ponder on when re-working the ‘Introduction section’. These are: what are the steps taken in order to ensure more comprehensive data collection from IMAM programs in Somalia? Has it been fruitful? Does IMAM programmes vary across different regions within Somalia, and if it does, what impact does this have on the outcomes in Somalia?
Response: We do disagree with this comment. As reviewer 2 notes “This is a good study; research question was well defined and relevant for the current practice. Though there are many publications in this area, authors have provided sufficient justification to show the knowledge gap in Somalia” and reviewer 3 adds “Introduction is well written and thorough”. A good introduction should answer three questions:
- What do we already know? This is comprehensively addressed from line 42 to 95, supported with key references
- What are the gaps? These are comprehensively addressed from Line 96 to 118
- As not all gaps are worth addressing (ie. We cannot address the world’s issues in one study), which of the identified gaps the study seeks to address and what are associated research questions? Thes are addressed from Line 119 to 127
Comment 5:
Methods: Provide a description of the study design - what is the type of design you employed in this study even though it is a systematic review and meta-analysis approach you employed. Is it cross-sectional approach in the collection of existing studies or in the longitudinal form.
Response: We are not sure whether this reviewer is familiar with the PRISMA guidelines. We clearly indicate that this study was conducted according to the preferred reporting items for systematic review and meta-analysis (Lines 129-130). Consistent with the PRISMA guidelines, the inclusion criteria clearly state the study was informed by the PICOTS framework (Population, Intervention, Comparison, Outcome, Timing, and Setting). We clearly show that the intervention is: IMAM, mainly OTP and SC cohorts, with the comparison being cohort monitoring through pre (admission)-post (discharge) analysis (Lines 150-151)
Comment 6
Give a detailed geographical description of the study setting and what is the justification for the choice of the study setting?
Response: The PICOT framework clearly states that the setting was Somalia (Lines 147-156), and the justification was in the rationale i.e. why we focused on Somalia (from Line 196)
Comment 7
Put the search strategy section in tables [143-144]
Response: We do not believe this constitute a table. Doing so will burden the paper with unnecessary Tables (as we already have many). We have, nonetheless, reformatted the text for clarity
Comment 8
Mention the total number of studies and participants included in the final analysis, if not already specified [173-186]
Response. Consistent with PRISMA guidelines, this was provided in the results from line 247 to 264, as well as Figure 1.
Comment 9:
Was there a concern for ethical considerations? If yes, what were ethical principles you employed when retrieving these studies for the systematic review and meta-analysis? Have it in mind that eethical considerations guide your research designs and practices, concerning consent, anonymity, confidentiality (if needed such as administrative documents), risk, and results communication (very important).
Response: We refer this reviewer to the PRISMA guidelines. A systematic review does not include primary data, but it uses publicly available data including but not public sources like scientific literature, government databases, clinical trial registries, and other openly available datasets. The reviewer may want to consult the following document: “The PRISMA 2020 statement: An updated guideline for reporting systematic reviews”, Plos Medicine https://pmc.ncbi.nlm.nih.gov/articles/PMC8007028/
Comment 10
Results: The thematic analysis is clear, but ensure each theme is introduced with a brief summary before diving into detailed findings [Line 226 - Line 314].
Response: Our paper is a meta-analysis, not a meta-ethnography. Not sure what the reviewer refers to as thematic analysis. The results are reported as per PRISMA guidelines [“ The PRISMA 2020 statement: An updated guideline for reporting systematic reviews”, Plos Medicine https://pmc.ncbi.nlm.nih.gov/articles/PMC8007028/ ] The operational definition of all concepts in the results is provided under Methods, section “ 2.5. Outcomes of interest”, Lines 200-231
Comment 11
Consider adding specific examples or case studies to illustrate the identified themes more vividly, in the areas of performance or under-performance of integrated management of acute malnutrition programmes in Somalia.
Response: Our study is not a meta-ethnography, but rather a clear meta-analysis for concepts clearly defines under section 2.5. It is very hard to follow this reviewer’s comment, but expert in this field should be able to follow the analyses and the flow of our narration. Not sure what is meant by case study as we did not collect primary data
Comment 12
How did you address the publication bias as mentioned in Line 311 – Line 313.
Response: Clearly stated from line 236 to 244 when we note “Egger’s tests were used for assessing publication bias and small-study effects (funnel plots and regression- and correlation-based methods)[35]. The Stata command “meta-bias” was used to test for funnel plot asymmetry. Egger's regressions were used to check for asymmetry in a funnel plot, regression-based methods are useful for publication bias, especially when detecting small-study effects [36, 37]. Funnel plots, weighted correlation, Egger’s regression bias coefficients, and regression scatter plots were produced for each study outcome (appendix 2). A non-statistically significant Egger’s regression bias coefficient provided weak evidence for the presence of small-study effects[36]” So, the reviewer’s comment is not clear, are they proposing a different method?
.
Comment 13
Discussion: The discussion section discusses the findings’ significance tied to the research objective(s), delving into the meaning, importance, and relevance of the study findings. Also, its focus on explaining and evaluating what you found must be related to the related to the existing literature review, making an argument in support or not supporting the overall argument. Therefore, let your discussion focused on the interpretation of your findings and relate to literature whether it corroborates or not, with existing studies, alongside with in-citation of recent references.
Response: Not sure we understand this comment. A good discussion should answer four questions: what was found? Does it support (replication) or refute (new findings) known knowledge? What is the public health significance of the findings? Can the findings be influenced by any methodological flaws That is what we have exactly done.
Comment 14
Some of these questions should come forward aligning with the study objectives you seek to fill the gap with: are there primary barriers to successfully implement IMAM programmes in Somalia, and, if no, how can this be addressed? How does the quality of care in OTPs and SCs affect the performance outcomes, and are there measures put in place for possible improvement? How do the performance outcomes of IMAM programs in Somalia compare with those in other countries facing similar challenges? Is there any trends or variations different from your findings? Can community engagement and local knowledge be integrated into the IMAM programmes to enhance their effectiveness? Any training and resources needed for healthcare providers to improve the management of acute malnutrition in Somalia? How can the long-term impact of IMAM programmes on child health and nutrition be measured and evaluated?
Response: The comparison with international standards and findings from other countries is clearly laid out in the first 3 paragraphs of the discussion. The evaluation of IMAM cannot deviate from the “Sphere Handbook: Humanitarian Charter and Minimum Standards in Humanitarian Response freely available here for reference https://www.spherestandards.org/handbook-2018/ The handbook outlines how IMAM program should be implemented and sets out minimum standards to guide the evaluation of such programs to allow comparison within and between countries. Lines 369-375 discuss some of the challenges associated with IMAM program implementation and uptake and the sub-sequent para explains why the success was evident in Somalia. We are hoping this reviewer read the paper
Comment 15
These will help tailor your discussion section more impactful and interesting, as these are some of the major concerns debated in High-income countries (HICs) and Low-and Middle-income countries (LMICs). Kindly revise.
Response: See response to comments 13 and 14
Comment 16
Study Implications for future studies/research: What are the implications of your study findings? What is the contribution of this study to the existing one, especially in sub-Saharan African or in developing countries?
Response: These were provided in Lines 376-407.
Comment 17
Regarding implications for future research, what role do national and international policies play in the success of IMAM programmes in Somalia, and how can these policies be strengthened to support IMAM programmes in Somalia?
Response: For example, from line377 to 392 we note “
“Our observed low relapse rate suggests good linkage and referral systems between OTPs, SCs and supplementary feeding programs, as well as optimal care practices and admission and exit criteria [6]. In addition, the humanitarian system has adapted well to the unique changing Somali context by supporting fixed and mobile OTPs[14, 15, 50]. Unfortunately, analysis by types of OTP (mobile vs fixed) programs was challenging due to incomplete aggregated data. Nonetheless, mobile OTPs allow the humanitarian response to reach people who are marginalized and excluded from accessing much-needed aid and treatment. First piloted in 2004 to respond to the needs of Somalis severely affected by drought and the measles epidemic, mobile health teams had neither formal training nor specific guide-lines to support their work[14, 15, 50]. Over the last decades, mobile health teams have evolved and expanded to mobile health and nutrition teams to include IMAM, sup-ported by training packages and guidelines that address three components: maternal and newborn care services (e.g., ante and post-natal services), child health (e.g. IMAM), and capacity building (e.g. support in disease outbreak response, health and nutrition service delivery, community mobilization and active case finding and referral, and system strengthening)[50]”. This is a clear policy adaptation in the Somali context, as it is the one country with a hybrid model incorporating fixed and mobile OTPs and SCs
Comment 18
Regarding implications for future studies, are there innovative approaches or technologies that can be incorporated into IMAM programmes to improve outcomes in Somalia?
Response: See response to comment 14. All IMAM programs have to adhere to the Sphere Guidelines (which have gone through a series of revisions over the years to adapt to the technological changes)
Comment 19
Strength of this study
What is the strengths of this study? You only included the Limitations for this study. This research should have its strengths in areas of literature review, methodological approach, and findings interpretations.
Response: Done. From Line 419
Comment 20:
Conclusion/Recommendations: The conclusion should stem from summing up your research paper (by the following steps - restate your research topic, restate the thesis, summarize the main points, state the significance or results, and conclude your thoughts). Always have it in mind that a conclusion is not merely a summary of the main topic(s) covered or a re-statement of your research problem, but a synthesis of key points and, if applicable, where you should recommend new areas for future research. Although, the conclusion is strong, consider reiterating the key implications for public health policy and future research. Where is the Recommendation for this study? You can add it to the Conclusion section with the sub heading – Conclusion/Recommendations
Response: We seem to be going in circle. This is a scientific report and not a government report that must outline prescriptive recommendations. The implementation of the IMAM programs is mandated by the Sphere guidelines, of which the ministries of health are signatories. The conclusion we provide is concise, specific to our findings and context, with international application. The specific recommendation we provide are outlined from line 428 ie. “The adaptation of IMAM programs to meet multiple challenges in Somalia, supported by well-designed, coordinated, standardized, integrated, and harmonized implementation plans, is a strength. International aid agencies and policymakers need to systematically document factors that have driven this success and assess how such lessons can be replicated and scaled up regionally”
Comment 21:
Editing: Ensure consistency in tense and improve grammatical flow in some sections
Response: Done
Comment 22. References
Does this References of this study follow this Journal format? Kindly check and revisit them all. Ensure all references are formatted correctly and consistently.
Response: We followed IJERPH guidelines. Any specific comment?
Reviewer 2 Report
Comments and Suggestions for Authors
This is a good study; research question was well defined and relevant for the current practice. Though there are many publications in this area, authors have provided sufficient justification to show the knowledge gap in Somalia.
Line 77-80 ; Better not to use terms marasmus and kwoshiokor, these are severe form conditions and does not tally with all SAM. Better to maintain WHO classification.
Line 137: Add the time frame for the search
Line 280: One of the authors from UNICEF Somalia, need to revise this.
Line 248: Need to include the bias assessment. UNICEF data were routinely collected data and the funder was also UNICEF, add few lines how the data quality is evaluated.
Research topic indicate this is a meta-analysis. Need to specify the appropriate method used for data combination. It is not clear how the heterogenicity was assessed - is it only random-effects model or whether sub group analysis also done? Was there any necessity to c onduct sensitivity analyses? Please check with your statistician.
Line 325: It is not clear the number of studies included. All theUNICEF dara were only presented. What about other 2 studies identified in the search.
This study has bit of conflict of interest. Majority of data is from UNICEF and authors are also from UNICEF and funding for all the data used in the review also from UNICEF. Paper is also supported by UNICFE. Need the strong declaration.
Gap in knowledge was not addressed using the study findings.
Need to discuss the positive and negative findings adequately in the results section and discuss the findings adequately.
Author Response
Comment 1:
This is a good study; research question was well defined and relevant for the current practice. Though there are many publications in this area, authors have provided sufficient justification to show the knowledge gap in Somalia.
Response: Thank you
Comment 2: Line 77-80; Better not to use terms marasmus and kwashiorkor, these are severe form conditions and does not tally with all SAM. Better to maintain WHO classification.
Response: First all, in the paragraph identified by this reviewer we identify three forms of severe acute malnutrition: marasmus (non-edematous), kwashiorkor (edematous) and marasmus and kwashiorkor (a combination of both), a classification which adheres to WHO classification. It is not clear what the reviewer is referring to when they state: “these are severe form conditions and does not tally with all SAM”. Although past literature conceived severe acute malnutrition to be a form of protein-energy deficiency, the emerging evidence has demystified this belief. A new classification has emerged, notably ‘type 1’ deficiency a condition in which the body continues to grow while depleting its nutrient stores, leading to noticeable symptoms as the body functions decline (i.e. there are clear signs to diagnose the disease as is the case for Kwashiorkor or marasmic kwashiorkor) and ‘type 2 deficiency’ a condition in which the body stops growing to conserve the nutrient, making it harder to identify the specific deficient nutrient due to a lack of distinct symptoms and hard to diagnose the cause (e.g marasmus). We encourage the reviewer to read: World Health Organization. Guidelines: updates on the management of severe acute malnutrition in infants and children. World Health Organization, 2013. Hence, no change was made.
Comment 3
Line 137: Add the time frame for the search
Response: Please see the PICOT framework, line 155
Comment 4
Line 280: One of the authors from UNICEF Somalia, need to revise this.
Response: we checked this and could still not understand what the reviewer meant. Data were extracted independent of UNICEF staff to remove any potential conflict of interest. The flowchart is very clear on steps taken
Comment 5:
Line 248: Need to include the bias assessment. UNICEF data were routinely collected data and the funder was also UNICEF, add few lines how the data quality is evaluated.
Response: Consistent with the PRISMA guidelines, the bias assessment is detailed in the data analysis section lines 237-244
Comment 6
Research topic indicates this is a meta-analysis. Need to specify the appropriate method used for data combination. It is not clear how the heterogenicity was assessed - is it only random-effects model or whether sub-group analysis also done? Was there any necessity to conduct sensitivity analyses? Please check with your statistician.
Response: Our aim was to show trends over time and then provide a pooled analysis. That is what we did. What does this reviewer mean by data combination? Do they mean merging data by year, which is what we did?
Comment 7
Line 325: It is not clear the number of studies included. All the UNICEF data were only presented. What about other 2 studies identified in the search.
Figure 1, and lines 217-260 are unequivocal in addressing this issue
Comment 8
This study has bit of conflict of interest. Majority of data is from UNICEF and authors are also from UNICEF and funding for all the data used in the review also from UNICEF. Paper is also supported by UNICFE. Need the strong declaration.
Response: Apart from their input into the research questions, the data analyses and interpretation of the results excluded all UNICEF staff. This his is reflected in the author’s contribution statement.
Gap in knowledge was not addressed using the study findings.
Need to discuss the positive and negative findings adequately in the results section and discuss the findings adequately.
Response: see lines 352 -407.
Reviewer 3 Report
Comments and Suggestions for Authors
Introduction is well written and thorough.
Line 87 misspelled center, and pediatric. Suggest checking spelling to convert to American English throughout the paper. Show consistency in language.
Line 129 extra space after "the".
Line 250 - check rules for capitalization, eg. "other"
Figure 1 - suggest consistency in formatting, eg. use or nonuse of semicolons
Figure 1 Line 267 misspelled "Excluded'"
Author Response
Introduction is well written and thorough.
Thank you
Line 87 misspelled center, and pediatric. Suggest checking spelling to convert to American English throughout the paper. Show consistency in language.
Response: American English or UK English are fine so long as there is consistency. We consistently used UK English. Thank you
Line 129 extra space after "the". Line 250 - check rules for capitalization, eg. "other". Figure 1 - suggest consistency in formatting, eg. use or nonuse of semicolons Figure 1 Line 267 misspelled "Excluded'"
Response: “other” does not need capital letter. “Excluded” corrected in figure 1. Formatted issues addressed.
Round 2
Reviewer 1 Report
Comments and Suggestions for Authors
The Authors address all comments.